# Cold Storage Media versus Optisol-GS in the Preservation of Corneal Quality for Keratoplasty: A Systematic Review

**Izabela Gimenes** [1,2], **Andréa V. Braga Pintor** [3], **Mariana da Silva Sardinha** [2], **Guido A. Marañón-Vásquez** [3], **Marcelo Salabert Gonzalez** [4], **Octavio Augusto França Presgrave** [2,5], **Lucianne Cople Maia** [3] **and Gutemberg Gomes Alves** [6,7,*]

1 Post-Graduation Program in Science and Biotechnology, Fluminense Federal University, Niterói 24020-140, Brazil; izabelagimenes@id.uff.br
2 National Institute for Quality Control in Health, Oswaldo Cruz Foundation, Rio de Janeiro 21040-900, Brazil; marianasardinhafiocruz@gmail.com (M.d.S.S.); octavio.presgrave@incqs.fiocruz.br (O.A.F.P.)
3 Department of Pediatric Dentistry and Orthodontics, Faculty of Dentistry, Universidade Federal do Rio de Janeiro, Rio de Janeiro 21941-971, Brazil; andrea_pintor@hotmail.com (A.V.B.P.); guido_amv@hotmail.com (G.A.M.-V.); maia_lc@odonto.ufrj.br (L.C.M.)
4 Department of General Biology, Fluminense Federal University, Niterói 24020-140, Brazil; msgonzalez@id.uff.br
5 Brazilian Centre for Validation of Alternative Methods (BraCVAM), Oswaldo Cruz Foundation, Rio de Janeiro 21040-900, Brazil
6 Cell and Molecular Biology Department, Institute of Biology, Fluminense Federal University, Niteroi 24020-140, Brazil
7 Clinical Research Unit, Antônio Pedro Hospital, Fluminense Federal University, Niteroi 24033-900, Brazil
* Correspondence: gutemberg_alves@id.uff.br

**Abstract:** Optisol-GS is the most widely used pharmaceutical composition to preserve corneas for transplantation. This systematic review investigated the effects of different cold corneal storage media (CCSM) compared with Optisol-GS on the quality of stored corneas. The literature was searched throughout May 2022 on six databases and grey literature. Studies including corneas (population) exposed to distinct cold storage media (exposure) and Optisol-GS (comparison) that reported qualitative and/or quantitative parameters of cornea quality (outcome) were included. Methodological quality was assessed using ToxRTool. From 4520 identified studies, fourteen were included according to the eligibility criteria, comprising 769 evaluated cornea samples comparing Optisol-GS with commercial and noncommercial media. All studies showed good methodological quality. Experimental times ranged from 1–28 days, mainly using 4 °C as the preservation temperature. Viable endothelial cell density (ECD) and endothelial cell morphology (EC) were the most assessed parameters. ECD results for Cornisol were higher than Optisol-GS in 10 days ($p = 0.049$) and favored Cornea Cold™ up to 4 weeks ($p < 0.05$), which also showed better qualitative results. While the standardization of test protocols could improve comparisons, evidence indicates that most CCSM present similar performances on cornea preservation for transplantation at seven days, while some formulations may increase preservation at extended times.

**Keywords:** Optisol-GS; cornea; culture media; organ preservation solutions; storage corneal medium; systematic review

## 1. Introduction

Corneal diseases, the second leading cause of reversible blindness worldwide, affect the young and active population leading to a significant economic and social loss [1]. Keratoplasty is one of the main solutions for corneal blindness, evolving from the full transplantation of a healthy donor cornea (full-thickness penetrating keratoplasty) to the selective replacement of diseased layers, including procedures such as superficial anterior lamellar keratoplasty (SALK), automated lamellar therapeutic keratoplasty (ALTK), deep

anterior lamellar keratoplasty (DALK), and Descemet stripping automated endothelial keratoplasty (DSAEK) [2]. However, the lack of available donors and eye banks capable of providing corneas in adequate numbers and quality for transplants is still a reality despite many efforts. Furthermore, limitations of the different national corneal transplantation programs worldwide may contribute to lower rates of donor cornea usage and a considerable shortage of corneal graft tissue constantly identified by global surveys [3].

In this context, biomedical advances on tissue engineering and preservation have contributed with methods that are essential for enhancing the quality of donor corneas, mainly intending to preserve the integrity of corneal endothelium, a critical factor for the successful long-term outcome of corneal transplantation [4]. Furthermore, longer storage times allow for increased flexibility with a reduced waste of donor tissue. Nonetheless, the preservation of human donor corneas is still limited to a maximum of 10–14 days by hypothermic storage, considered a simple and effective method that uses cold cornea storage media [5]. Different studies have identified corneal storage for periods superior to 7 days as a risk factor for primary graft failure and lower 3-year graft survival after keratoplasty [6]. However, improvements in surgical techniques and preservation of cornea have reduced intraoperative cell loss and graft dislocation after procedures such as Descemet stripping automated endothelial keratoplasty (DSAEK), with clinically insignificant differences for cornea grafts under 12 days of preservation [6]. An Indian report in 2019 indicates that the storage medium is a major factor among the various causes that affect the success rates of donated cornea utilization, which may reach 88% in developed countries with access to high-quality products [7].

Nowadays, different types of cold cornea storage media are used in corneal preservation for different purposes, such as corneal transplantation. Optisol (Bausch and Lomb Inc., Rochester, NY, USA), a hybrid of K-sol and Dexsol, was introduced in the early 1990s [5]. The addition of two different antibiotics, gentamicin sulfate and streptomycin sulfate, resulted in the development of Optisol-GS [5,8–10]. This distinct pharmaceutical composition is used as a tissue culture medium, enriched with polypeptides, dextran (osmotic agent), chondroitin sulfate, and the aforementioned antibiotics. This commercial solution improved antimicrobial efficacy, even though there is still no hypothermic antifungal agent commercially available in the USA [5]. Optisol-GS is traditionally considered as the popular choice among storage media used in the United States and is the "gold standard" medium for preservation at 2–8 °C for up to 14 days before corneal transplantation [11]. However, regarding the cost perspective, Optisol-GS can be considered expensive and less affordable. Surandesan et al. [12] pointed out that reasonably priced and effective storage media are an integral part of a successful cost-effective corneal transplantation program, especially in developing countries, leading to the need for a wider choice of available options for the international ophthalmological community. In this sense, advances in corneal preservation for keratoplasty are achieved by the development and proposal of novel formulations of cold storage media aimed at improving cost-effective corneal transplantation care [11].

While several storage media such as Optisol, Eusol, Cornisol, and Life 4 °C are considered safe and effective, having their use approved by regulatory agencies such as the American Food and Drugs Administration (FDA), these products do not necessarily present the same performances and qualities of preservation, especially regarding longer preservation times. In this sense, several studies have investigated the effects of different culture media on the quality of stored corneas, employing different qualitative and quantitative parameters [13]. However, there is no comprehensive and systematic comparison between the effects of the different culture media on corneal preservation that gathers important parameters of corneal quality assessment, such as corneal transparency (CT) and endothelial cells (EC) morphology; quantitative parameters, e.g., endothelial cell density (ECD); central corneal thickness (CCT); and EC mortality. Therefore, this study aimed to systematically review the literature regarding the effects of the use of different cold corneal storage media compared with the use of Optisol-GS on the quality of stored corneas and to answer the

PECO focused research question: what are the effects of different cold preservation media versus Optisol-GS on corneal quality preservation?

This review assessed the main characteristics of the different novel storage media and the relationship between their chemical formulation and performance compared with the up-to-date gold standard in transplantation and discussed the relevance of standardization of the available parameters for the quality of stored corneas. The comparison of these different solutions may fill a gap in the literature on keratoplasty regarding the choice of inputs for cornea preservation and contribute to the scientific knowledge of decision makers in this critical step of corneal transplantation and eye bank management.

The remainder of the paper is structured as follows. The methodology implemented to conduct this systematic review is described in detail in Section 2, while the results are reported in Section 3 following the Preferred Reporting Items for Systematic Reviews and Meta-Analyses (PRISMA) [12]. Section 4 shows a narrative discussion of the literature findings. Lastly, Section 5 concludes the paper.

## 2. Materials and Methods

### 2.1. Protocol and Registration

This systematic review was reported following the recommendations of the Preferred Reporting Items for Systematic Reviews and Meta-Analyses (PRISMA). It consists of a 27-item checklist with sections and subsections recommended for reporting in the present systematic review (Supplementary Table S1) [14]. The study protocol was registered in the Open Science Framework Database, available at the following link: osf.io/qh69k/, accessed on 24 August 2021.

### 2.2. Research Question and Eligibility Criteria

Considering that there is no broad comparison between distinct cold cornea storage media effects on corneal preservation, the outline of the main research question was motivated by four different questions to address the gap of the literature, as follows: population (P): which types of corneas were investigated?; exposure (E): which methods of corneal preservation should be considered?; comparators (C): Is there a "gold-standard" corneal cold preservation medium?; outcome (O): which were the cornea preservation quality parameters? As a result, following the PECO framework (Table 1), the main research question was formulated as "What are the effects of different cold preservation media versus Optisol-GS on corneal quality preservation?"

**Table 1.** PECO framework.

| PECO Framework | |
|---|---|
| **P** **Population** | Human or animal corneas |
| **E** **Exposure** | Distinct cold storage corneal medium |
| **C** **Comparator** | Optisol-GS |
| **O** **Outcomes** | Evaluation of qualitative parameters: corneal transparency (CT) and endothelial cells (EC) morphology; quantitative parameters: endothelial cell density (ECD), central corneal thickness (CCT), and EC mortality |

Similarly, the eligibility criteria that guided the selection process of the studies included were based on the PECO structure. The inclusion criteria comprises in vitro and ex vivo studies conducted on human or animal corneas, which compared the use of distinct cold storage corneal medium to the use of Optisol-GS regarding corneal quality preservation parameters. The exclusion criteria comprised studies that reported data after cornea transplantation, case reports, reviews, observational studies, letters to the editors, editorials, commentaries, conference abstracts, and book chapters that did not present primary data.

In addition, studies that were not conducted on cornea and those that did not include Optisol-GS as a comparative group for the evaluation of corneal quality preservation were excluded. There was no year/time constraint for the selected studies.

### 2.3. Information Sources

The electronic search was performed in June 2021 and updated in May 2022, on PubMed (http://www.ncbi.nlm.nih.gov/sites/pubmed, accessed on 21 June 2021); Scopus (http://www.scopus.com, accessed on 21 June 2021, accessed through Advanced Search, Enter query string); Embase (https://www.embase.com, accessed on 22 June 2021), accessed through Elsevier (https://www.elsevier.com, accessed on 22 June 2021); Web of Science, WOS (https://www.webofknowledge.com, accessed on 21 June 2021), accessed through the Clarivate Analytics (https://clarivate.com, accessed on 22 June 2021); Cochrane Library (https://www.cochranelibrary.com, accessed on 22 June 2021); and LILACS database was consulted in Virtual Health Library, VHL (https://bvsalud.org, accessed on 22 June 2021). Grey literature was consulted through OpenGrey (www.opengrey.eu, accessed on 6 August 2021) and Google Scholar (https://scholar.google.com, accessed on 6 August 2021.). Database alerts were set to retrieve new publications. Experts in the field were identified at Expert Scape (https://www.expertscape.com/, accessed on 4 January 2022) and contacted by the "List Experts" for ongoing studies or unpublished results regarding the focused question, using up to five email contact attempts until January 2022.

### 2.4. Search Strategy

Medical Subject Headings (MeSH) terms (www.nlm.nih.gov/mesh/meshhome.html, accessed on 21 June 2021), entry terms, free terms, and keywords related to the aim of this review were included in the search strategy. No restrictions on language or date were applied. The search strategy was developed using the Boolean operators AND/OR for PubMed and then adapted to each database according to their syntax rules. A manual search was carried out in the reference lists of the articles selected for the systematic review to detect relevant publications missed in the database searches. Documents from Google Scholar covered the first 200 matches, which were manually processed to verify whether possible eligible studies were missing from the primary database search. Records retrieved from more than one database were computed only once. Authors and co-authors of studies not retrieved in full text were contacted by email (up to five attempts), from October to December 2021. The search strategy is described in Table 2.

**Table 2.** Search Strategy.

| Database | Search Strategy |
|---|---|
| **PUBMED** | (Cornea[Mesh] OR Cornea*[tiab] OR cornea ex vivo[tiab]) AND (Culture Media[Mesh] OR HEPES[Mesh] OR Cryoprotective Agents[Mesh] OR Organ Preservation solutions[Mesh] OR Organ Preservation[Mesh] OR Cryopreservation[Mesh] OR glucose[Mesh] OR acids[Mesh] OR vitamins[Mesh] OR penicillins[Mesh] OR streptomycin[Mesh] OR Bicarbonates[Mesh] OR Adenosine Triphosphate[Mesh] OR Methylcellulose[Mesh] OR Culture Media[tiab] OR Cryopreservation[tiab] OR Adenosine Triphosphate[tiab] OR cornea max[tiab] OR cold storage medium[tiab] OR hypothermic storage[tiab]) AND (Chondroitin Sulfates[Mesh] OR Dextrans[Mesh] OR Gentamicins[Mesh] OR Complex Mixtures[Mesh] OR Optisol[tiab] OR Dextran[tiab] OR Gentamicin Sulfate[tiab]) |
| **SCOPUS** | INDEXTERMS(Cornea) OR TITLE-ABS-KEY(Cornea*) OR TITLE-ABS-KEY("cornea ex vivo") AND INDEXTERMS("Culture Media") OR INDEXTERMS(HEPES) OR INDEXTERMS("Cryoprotective Agents") OR INDEXTERMS("Organ Preservation solutions") OR INDEXTERMS("Organ Preservation") OR INDEXTERMS(Cryopreservation) OR INDEXTERMS(glucose) OR INDEXTERMS(acids) OR INDEXTERMS(vitamins) OR INDEXTERMS(penicillins) OR INDEXTERMS(streptomycin) OR INDEXTERMS(Bicarbonates) OR INDEXTERMS("Adenosine Triphosphate") OR INDEXTERMS(Methylcellulose) OR TITLE-ABS-KEY("Culture Media") OR TITLE-ABS-KEY(Cryopreservation) OR TITLE-ABS-KEY("Adenosine Triphosphate") OR TITLE-ABS-KEY("cornea max") OR TITLE-ABS-KEY("cold storage medium") OR TITLE-ABS-KEY(" hypothermic storage") AND INDEXTERMS("Chondroitin Sulfates") OR |

**Table 2.** *Cont.*

| Database | Search Strategy |
|---|---|
| | INDEXTERMS(Dextrans) OR INDEXTERMS(Gentamicins) OR INDEXTERMS("Complex Mixtures") OR TITLE-ABS-KEY(Optisol) OR TITLE-ABS-KEY(Dextran) OR TITLE-ABS-KEY("Gentamicin Sulfate") |
| **WOS** | TS = (Cornea) OR TS = (Corneas) OR TS = (corneal) OR TS = ("cornea ex vivo") AND TS = ("Culture Media") OR TS = (HEPES) OR TS = ("Cryoprotective Agents") OR TS = ("Organ Preservation solutions") OR TS = ("Organ Preservation") OR TS = (Cryopreservation) OR TS = (glucose) OR TS = (acids) OR TS = (vitamins) OR TS = (penicillins) OR TS = (streptomycin) OR TS = (Bicarbonates) OR TS = ("Adenosine Triphosphate") OR TS = (Methylcellulose) OR TS = ("cornea max") OR TS = ("cold storage medium") OR TS = ("hypothermic storage") AND TS = ("Chondroitin Sulfates") OR TS = (Dextrans) OR TS = (Gentamicins) OR TS = ("Complex Mixtures") OR TS = (Optisol) OR TS = (Dextran) OR TS = ("Gentamicin Sulfate") |
| **COCHRANE** | #1 MeSH descriptor: [Cornea] explode all trees<br>#2 (Cornea* OR cornea ex vivo):ti,ab,kw<br>#3 = #1 OR #2<br>#4 MeSH descriptor: [Culture Media] explode all trees<br>#5 MeSH descriptor: [HEPES] explode all trees<br>#6 MeSH descriptor: [Cryoprotective Agents] explode all trees<br>#7 MeSH descriptor: [Organ Preservation Solutions] explode all<br>#8 MeSH descriptor: [Organ Preservation] explode all trees<br>#9 MeSH descriptor: [Cryopreservation] explode all trees<br>#10 MeSH descriptor: [Glucose] explode all trees<br>#11 MeSH descriptor: [Acids] explode all trees<br>#12 MeSH descriptor: [Vitamins] explode all trees<br>#13 MeSH descriptor: [Penicillins] explode all trees<br>#14 MeSH descriptor: [Streptomycin] explode all trees<br>#15 MeSH descriptor: [Bicarbonates] explode all trees<br>#16 MeSH descriptor: [Adenosine Triphosphate] explode all trees<br>#17 MeSH descriptor: [Methylcellulose] explode all trees<br>#18 (Culture Media OR Cryopreservation OR Adenosine Triphosphate OR cornea max OR cold storage medium OR hypothermic storage):ti,ab,kw<br>#19 = #4 OR #5 OR #6 OR #7 OR #8 OR #9 OR #10 OR #11 OR #12 OR #13 OR #14 OR #15 #16 OR #17 OR #18<br>#20 MeSH descriptor: [Chondroitin Sulfates] explode all trees<br>#21 MeSH descriptor: [Dextrans] explode all trees<br>#22 MeSH descriptor: [Gentamicins] explode all trees<br>#23 MeSH descriptor: [Complex Mixtures] explode all trees<br>#24 (Optisol OR Dextran OR Gentamicin Sulfate):ti,ab,kw<br>#25 = #3 AND #19 AND #24 |
| **EMBASE** | 'cornea'/exp OR cornea*:ti,ab,kw OR 'cornea ex vivo':ti,ab,kw AND 'culture media'/exp OR 'hepes'/exp OR 'cryoprotective agents'/exp OR 'organ preservation solutions'/exp OR 'organ preservation'/exp OR 'cryopreservation'/exp OR 'glucose'/exp OR 'acids'/exp OR 'vitamins'/exp OR 'penicillins'/exp OR 'streptomycin'/exp OR 'bicarbonates'/exp OR 'adenosine triphosphate'/exp OR 'methylcellulose'/exp OR 'culture media':ti,ab,kw OR 'cryopreservation':ti,ab,kw OR 'adenosine triphosphate':ti,ab,kw OR 'cornea max':ti,ab,kw OR 'cold storage medium':ti,ab,kw OR 'hypothermic storage':ti,ab,kw AND 'chondroitin sulfates'/exp OR 'dextrans'/exp OR 'gentamicins'/exp OR 'complex mixtures'/exp OR 'optisol':ti,ab,kw 'dextrans':ti,ab,kw OR 'gentamicin sulfate':ti,ab,kw |
| **LILACS via VHL** | (mh:cornea OR cornea* OR "cornea ex vivo") AND (mh:hepes OR mh:"culture media" OR mh:"cryoprotective agents" OR mh:"organ preservation solutions" OR mh:"organ preservation" OR mh:cryopreservation OR mh:glucose OR mh:acids OR mh:vitamins OR mh:penicillins OR mh:streptomycin OR mh:bicarbonates OR mh:"adenosine triphosphate" OR mh:methylcellulose OR "culture media" OR cryopreservation OR "adenosine triphosphate" OR "cornea max" OR "cold storage medium" OR "hypothermic storage") AND (mh:dextrans OR mh:"chondroitin sulfates" OR mh:gentamicins OR mh:"complex mixtures" OR optisol OR dextran OR "gentamicin sulfate") |
| **OPENGREY** | Cornea and Optisol |
| **GOOGLE SCHOLAR** | Cornea and Optisol |

* was used to retrieve both the singular and plural forms.

*2.5. Selection Process*

The retrieved documents were exported and organized in the Rayyan[TM] Web and Mobile App for Systematic Reviews (https://www.rayyan.ai/, accessed on 22 June 2021) reference management software. Duplicates were removed manually and checked by a reviewer (I.G.L) for the remaining ones. Two reviewers (I.G.L.; A.V.B.P.) read the titles and abstracts independently, using the blinded process on Rayyan® Web to determine whether the articles met the inclusion and exclusion criteria according to the PECO strategy. Selected articles were fully read to confirm eligibility. If the two reviewers became unsure about the inclusion/exclusion of any article, the issue was solved through a consensus meeting with a third senior reviewer (G.G.A.). Reasons for exclusion of articles after full-text examination were registered. Articles published in languages that the authors of this study did not know were translated using the Google[TM] Translate Tool at https://translate.google.com, accessed on 25 November 2021.

*2.6. Data Collection Process and Data Items*

Two reviewers (I.G.L; M.S.) independently performed the data extraction manually. The primary study characteristics were identified and organized on Excel spreadsheets (Excel 2010®, Microsoft®, USA), including first author, year of publication, the country where the research was conducted, study design, sample type, sample size, eye bank, distinct exposition media, storage time, storage temperature, assessment/methods of analysis, qualitative and quantitative parameters for cornea preservation after cold storage in Optisol-GS. Reported cornea quality parameters, main results, and conclusions were also collected from the included studies. The outcomes of interest were reported exclusively for the storage period at single or multiple experimental times.

*2.7. Study Risk of Bias Assessment*

Two independent reviewers (I.G.L.; A.V.B.P.) assessed the methodological quality of the included studies in compliance with the ToxRTool criteria (Toxicological data Reliability Assessment Tool) [15]. In case of doubts, a third reviewer would mediate (G.G.A.). Whenever necessary, up to five attempts of contact with the corresponding author were performed by email to retrieve any possible missing data in the included studies. ToxRTool for in vitro studies consists of an 18-point rating checklist, considering methodological aspects of each study, such as identification of test substance and test system, study design, and result documentation. Articles with less than 11 points are considered unreliable, while studies with 11–14 points are reliable with possible restrictions, and studies with 15–18 points are considered reliable without restrictions.

*2.8. Effect Measures and Synthesis Methods*

The characteristics of the included studies were summarized and tabulated using Excel spreadsheets (Excel 2010®, Microsoft, USA). The studies were grouped for the synthesis based on the outcomes of interest related to the qualitative and quantitative parameters of corneal preservation. Subsequently, the characteristics of the studies were screened to determine which were similar enough to be grouped within each comparison, exploring and comparing the PECO elements across the studies. Data were analyzed and interpreted qualitatively to integrate the reported information and present the synthesis results descriptively.

## 3. Results

The results of the study selection, quality assessment, study characteristics, as well as the results of synthesis are reported below in accordance with the provisions of the PRISMA Statement.

*3.1. Study Selection*

The initial search identified 4361 records and 159 were retrieved in the updated search in May 2022 as shown in Figure 1. After the removal of duplicates, a total of

4053 records were screened. Considering the eligibility criteria, 3969 records were excluded and 84 studies were selected for full-text reading. Four reports were not retrieved in full text [16–19] after unsuccessful contacts with authors or library sites. Considering 80 eligible studies, 66 were excluded for the following reasons: cells only (*n* = 7), no baseline (*n* = 6), no cold corneal storage media (*n* = 5), no Optisol-GS (*n* = 13), parts of cornea (*n* = 18), wrong outcomes (*n* = 12), wrong study design (*n* = 5), and data not retrieved for analysis (*n* = 1). In this manner, fourteen records originating from the main databases were selected. Regarding the records identified via grey literature, the first two hundred matches from the 3250 results in Google Scholar were assessed for the study. One duplicate record was manually removed from these sources, another ninety-seven were identified as duplicates by the Rayyan® Web software, and one was removed by duplicate records from Google. Furthermore, 92 records were excluded by title/abstract reading: no use of Optisol-GS (*n* = 22), wrong outcomes (*n* = 48), wrong study design (*n* = 21), and data not retrieved for analysis (*n* = 1). No remaining records were assessed for eligibility from the other resources since no articles were found on OpenGrey. Therefore, 14 studies were included in this review.

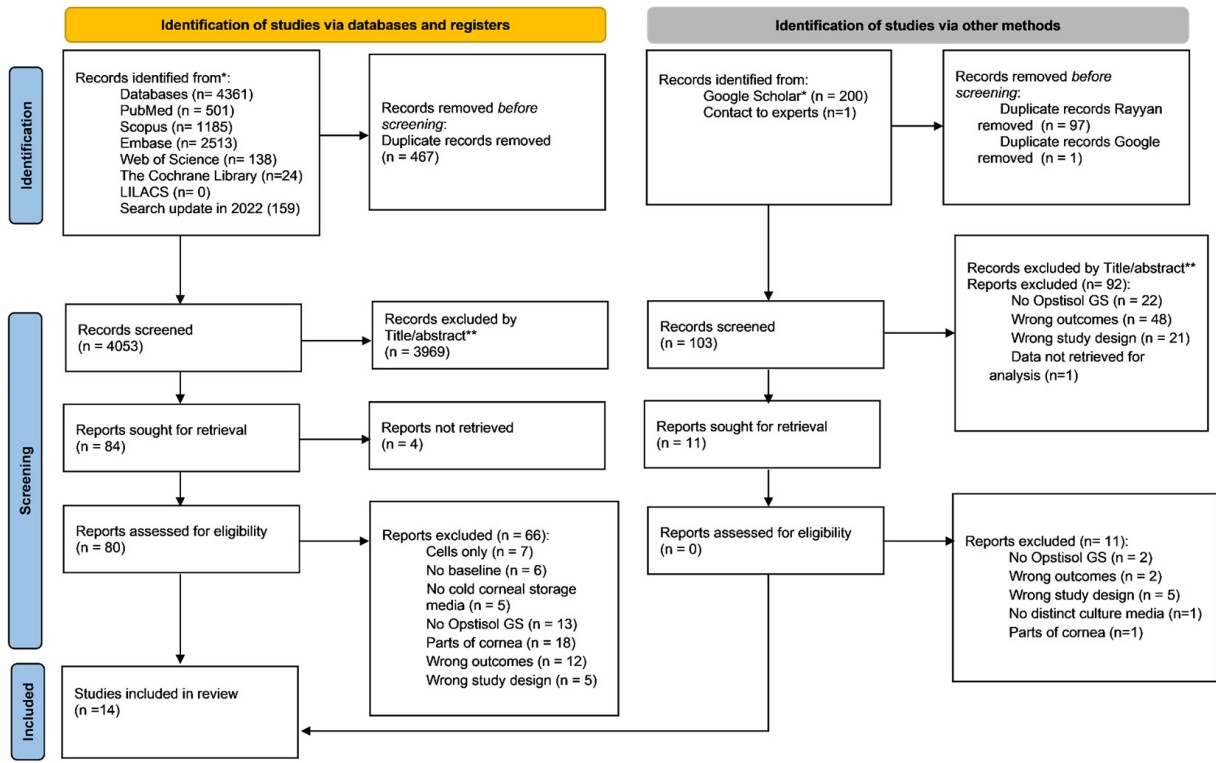

**Figure 1.** PRISMA 2020 flow diagram of the screening and selection process.

Most articles included were published in English, except for one in Japanese [20], which was translated with Google Translate. Experts identified in the Expertscape database by the "list experts" were contacted about unpublished results or material in progress: Parekh, M., Mehta, J., Jod S., Alió, J. L., Sharma, N., Dana, R., Kymionis, G. D., and Vardhaman, P. K. From those, Dr. Jod, Dr. Mehta, and Dr. Dana answered the email contact reporting not having any new data of interest to this systematic review. Dr. Namrata Sharma shared a recent article [21]; there was unsuccessful contact with the other authors from December 2021 to January 2022.

### 3.2. Quality Assessment

The methodological quality assessment of the included studies is reported in Table 3. All studies were considered "reliable without restriction" with good methodological quality according to ToxRTool criteria [15].

**Table 3.** Quality assessment according to the ToxRTool criteria.

| Reference | Group I: Test Substance Identification | Group II: Test System Characterization | Group III: Study Design Description | Group IV: Study Results Documentation | Group V: Plausibility of Study Design and Data | Total | Reliability Categorization |
|---|---|---|---|---|---|---|---|
| Basak and Prajna, 2016 [22] | 4 | 3 | 6 | 3 | 2 | 18 | Reliable without restrictions |
| Cid et al., 2021 [23] | 4 | 3 | 6 | 3 | 2 | 18 | Reliable without restrictions |
| Duncan et al., 2016 [24] | 4 | 3 | 6 | 3 | 2 | 18 | Reliable without restrictions |
| Greenbaum et al., 2004 [25] | 4 | 3 | 7 | 1 | 1 | 16 | Reliable without restrictions |
| Javadi et al., 2021 [26] | 4 | 4 | 6 | 3 | 2 | 19 | Reliable without restrictions |
| Kanavi et al., 2015 [27] | 4 | 3 | 6 | 3 | 2 | 18 | Reliable without restrictions |
| Layer et al., 2014 [28] | 4 | 3 | 6 | 2 | 2. | 17 | Reliable without restrictions |
| Li et al., 2012 [29] | 4 | 3 | 6 | 3 | 2 | 18 | Reliable without restrictions |
| Mistò et al., 2020 [30] | 4 | 3 | 6 | 3 | 2 | 18 | Reliable without restrictions |
| Nelson et al., 2000 [31] | 4 | 3 | 6 | 3 | 2 | 18 | Reliable without restrictions |
| Parekh et al., 2014 [13] | 4 | 3 | 6 | 3 | 2 | 18 | Reliable without restrictions |
| Perry et al., 2020 [32] | 4 | 3 | 6 | 3 | 2 | 18 | Reliable without restrictions |
| Smith et al., 1995 [8] | 4 | 3 | 6 | 3 | 2 | 18 | Reliable without restrictions |
| Tachibana and Sawa, 2001 [20] | 4 | 3 | 6 | 2 | 2 | 17 | Reliable without restrictions |

### 3.3. Study Characteristics

A total of 769 corneas were evaluated regarding the qualitative and quantitative parameters of tissue preservation. The main characteristics of the included studies, such as storage conditions, time and temperature ranges, the comparative distinct exposition media, and the assessment methods are shown in Table 4. Of the 14 included studies, 13 evaluated the effect of preservation in human corneas [5,12,22–32]; 1 study was performed with rabbit cornea for a preclinical evaluation of new intermediate storage media with simple formulations [8].

Optisol-GS was compared with seven distinct commercial media: Chen medium [31], Cornea Cold medium [13], Cornisol [22], Dexsol [25], Eusol-C [27], Kerasave [30,32], and Sinasol [26]. Optisol-GS was also compared with other compositions, such as a coconut water-based solution [23], Optisol-GS supplemented with antibiotics [24,28,29,33], and a minimum essential medium (MEM) with supplements [33].

The main components of the included cold cornea storage media are described in Tables 5 and 6. The most used components included: MEM (base medium) [13,22,25–27,30,32], HEPES (buffer) [13,24,26,27,31], and the antibiotics gentamicin and streptomycin [13,22,26,30–32]. The cell metabolism stimulant and hyperosmotic agent chondroitin sulfate is present in Optisol-GS, Dexsol [25], Sinasol [26], and other compositions [20]. All commercial media had Dextran as a hyperosmotic agent. ATP precursors were included as a source of intracellular energy in Optisol-GS, Cornisol [22], Cornea Cold medium [15], Eusol-C [34], and Dexsol [25]. Sodium pyruvate was also included as an energy source in Optisol-GS, Cornea Cold medium [13,35], Eusol-C [34], and Dexsol [25]. The use of sodium bicarbonate buffer was described for Optisol-GS, Kerasave [30,32], Cornisol [22], Cornea Cold medium (35), Eusol-C [27], Dexsol [25], and Sinasol [26]. Ascorbic acid was reported as an antioxidant agent in the composition of Optisol-GS.

Regarding the corneal preservation, experimental times ranged from 1 to 28 days. The most investigated experimental times were 7 days [13,20–26,28–31,34], and 14 days [13,22,24,26,28,30,34]. The mainly reported hypothermic temperature was 4 °C [8,24–27,29,31–33] (Table 4).

**Table 4.** Main characteristics of the selected studies.

| Author Year | Country | Eye Bank/Source | Sample Type | Exposition Media | Sample Size (*n*) | Storage and Assessment Period (Days) | Storage Temperature | Parameters (Assessment Methods) | Study Conclusion |
|---|---|---|---|---|---|---|---|---|---|
| Basak and Prajna, 2016 [22] | India | Rotary Aravind Eye Bank, Madurai; Aravind Eye Bank, Coimbatore; and Prova Eye Bank, Barrackpore. | Human corneas | Cornisol | 64 | 3, 7, 10, 14 | 2–8 °C | Corneal structure grading (slit lamp and specular microscope/histopathology); endothelial cell vitality (alizarin red S and trypan blue); ECD and ECL (specular microscopy) | This study concludes that CS is as effective as OS for storing corneal tissues at 2–8 °C for 14 days. |
| Cid et al., 2021 [23] | Brazil | Eye Bank of the General Hospital of Fortaleza. | Human corneas | Coconut water-based solution | 28 | 0, 1, 3, 7 | 2–8 °C | Viability: osmolarity (specular microscopy and slit lamp biomicroscopy) | Coconut water-based preservative partially maintained corneal transparency and epithelial integrity, especially during the first three days of follow-up. The coconut water-based solutions used were not effective for use as preservatives in a human eye bank. |
| Duncan et al., 2016 [24] | USA | SightLife | Human Corneas | Optisol-GS with amphotericin B | 24 | 0, 7, 14 | 4 °C | Endothelial cell density—ECD (specular microscopy); percentage of intact epithelium (slit lamp); damaged or dead endothelial cells (trypan blue); cell borders and areas of denuded Descemet membrane (alizarin red) | This study confirmed the efficacy and safety of amphotericin B as an antifungal supplement in Optisol-GS. Although all concentrations of amphotericin B effectively eliminated fungal contaminants within 7 days, only the 0.255-µg/mL concentration eliminated all fungal contaminants within 2 days. |
| Greenbaum et al., 2004 [25] | Canada | Eye Bank of Canada (Ontario Division) Toronto, Canada | Human corneas | Dexsol | 24 | 1, 2, 4 | 4 °C | Loss of donor epithelium (light microscopy); epithelium damage: basement membrane, cellular integrity, intercellular junctions, and intracellular organelles (transmission electron microscopy) | Loss of donor epithelium is related mainly to the length of storage and is similar in both Optisol GS and Dexsol. The storage time should be less than 4 days. |

**Table 4.** *Cont.*

| Author Year | Country | Eye Bank/Source | Sample Type | Exposition Media | Sample Size (*n*) | Storage and Assessment Period (Days) | Storage Temperature | Parameters (Assessment Methods) | Study Conclusion |
|---|---|---|---|---|---|---|---|---|---|
| Javadi et al., 2021 [26] | Iran | Central Eye Bank of Iran | Human corneas | Sinasol | 128 corneas (7 d); 59 corneas (14 d) | 7, 14 | 4 °C | ECD (specular microscopic) viability of the ECs (trypan blue staining) | The overall results indicate that Sinasol is a safe, effective, and affordable intermediate cold storage medium for preservation of corneas. |
| Kanavi et al., 2015 [27] | Iran | Central Eye Bank of Iran (CEBI) | Human corneas | Eusol | 180 | 1, 7 | 4 °C | Epithelial defects, stromal edema, Descemet's folding and an endothelial rating (slit lamp biomicroscopy) ECD (specular microscopic) | There was no significant change between Optisol-GS and Eusol-C and none of them seem to be superior to another (day 1 to day 7). |
| Layer et al., 2014 [28] | USA | SightLife | Human corneas | OGS with amphotericin B or voriconazole | 30 | 0, 7, 14 | 2–8 °C | Change in epithelium (slit lamp); endothelial cell count ECC (specular microscopy); vital dye staining (trypan blue) | A low concentration of amphotericin B might be a safe and efficacious addition to storage media, and a larger study is warranted to confirm these findings. No difference in the percentage of nonviable endothelial cells between paired controls and antifungal-supplemented Optisol-GS. |
| Li et al., 2012 [29] | USA | Eye Bank Association of America | Human Corneas | OGS with linezolid; daptomycin; calcium | 10 | 0, 7 ECD 0, 10 DCT | 4 °C | ECD (specular microscopy); DCT (ultrasound pachymetry) | The addition of daptomycin to Optisol-GS significantly increases the anti-MRSA activity of the medium without any apparent negative effects on donor corneal tissue. |
| Mistò et al., 2020 [30] | Italy | Eye Bank of Monza | Human corneas | Kerasave with amphotericin B and Optisol. | 32 | 1, 7, 14 | 2–8 °C | CCT, ECD (Azul trypan, specular microscopy), corneal transparency, EC morphology (slit lamp biomicroscopy) and inverted-phase microscopy. | No differences were found in the qualitative (corneal transparency, EC morphology), and quantitative metrics. |
| Nelson et al., 2000 [31] | USA | Eye Bank Specular Microscope | Human corneas | Chen Medium | 18 | 7, 10, 14, 21 | 4 °C | CT, ECD (specular microscopy), scanning electron microscopy (SEM), | Corneas stored in CM were thicker during storage than those stored in OM. |

**Table 4.** *Cont.*

| Author Year | Country | Eye Bank/Source | Sample Type | Exposition Media | Sample Size (*n*) | Storage and Assessment Period (Days) | Storage Temperature | Parameters (Assessment Methods) | Study Conclusion |
|---|---|---|---|---|---|---|---|---|---|
| Parekh et al., 2014 [13] | Italy | The Veneto Eye Bank Foundation | Human corneas | Cornea Cold | 60 | 7, 14, 21, 28 | 2–6 °C | CT, ECD, and morphology (optical microscopic evaluation); transparency (specific device) | Cornea Cold is a promising hypothermic corneal storage medium with preservation time until 21 days. |
| Perry et al., 2020 [32] | USA | Sight Eye Bank | Human corneas | Kerasave | 88 | 12 | 4 °C | ECD, CCT (slit lamp; specular microscope) | Kerasave should notice little difference when compared with Optisol-GS. This was not statistically significant. |
| Smith et al., 1995 [8] | USA | Los Angeles Doheny Eye Bank | Human corneas | OGS with gentamicin and streptomycin | 4 | 5 | 4 °C | Cytotoxicity; morphological or cell changes (SEM) | The shape and boundaries of the EC of each donor pair appeared to be similar. The quality of EC was poor, many cells were shrunken, and cytoplasm was separated from the plasma membranes, which were thick and irregular. |
| Tachibana and Sawa, 2002 [20] | Japan | - | Rabbit corneas | Medium with 2.5% chondroitin sulfate in different molecular weights: Medium I and II | 12 | 7, 14 | 4 °C | Cell morphology (SEM and TEM) | No significant difference in histological findings between Optisol-GS and test medium at days 5 and 10. On day 14, corneal endothelial cells with marked degeneration of intracellular organelles in both media at day 14. |

DCT—donor corneal thickness; ECC—endothelial cell count; CCT—central corneal thickness; SEM—scanning electron microscopy; TEM—transmission electron microscopy.

**Table 5.** List of commercial preservation media and their compositions.

| Components | Optisol-GS | Kerasave [24,33] | Cornisol [21] | Cornea Cold Medium [11] | Eusol-C [36] | Chen Medium [20] | Dexsol [22] | Sinasol [26] |
|---|---|---|---|---|---|---|---|---|
| Base medium | Tissue culture medium 199, Eagle's balance salt solution and MEM | MEM | MEM | MEM | MEM-Earle | Modified medium 199 | MEM | MEM |
| Buffer | HEPES | Sodium bicarbonate | HEPES | HEPES | HEPES | HEPES | HEPES | HEPES |
| Antibiotics | Gentamicin, streptomicin | gentamicin, streptomicin | Gentamicin, streptomicin | Yes | Gentamicin | Gentamicin, streptomicin | Gentamicin | Gentamicin, penicillin, streptomicin |
| Chondroitin sulfate | 2.5% | No | Yes | No | No | No | 1.35% | Yes |
| Dextran | T-40 | Yes | T-40 | T-500 | T-500 | Yes | T-40 | T-70 |
| ATP precursors | Yes | No | Yes | Yes | Yes | No | Yes | No |
| Sodium bicarbonate | Yes | Yes | Yes | Yes | Yes | No | Yes | Yes |
| Ascorbic acid | Yes | No | No | No | No | No | No | No |
| Sodium pyruvate | Yes | No | No | Yes | Yes | No | Yes | No |
| Additional supplements | Yes | No | No | Yes | Yes | Yes | Yes | No |

MEM—minimum essential medium, HEPES—N-2-hydroxyethylpiperazine-N-2 ethanesulfonic acid, ATP—adenosine triphosphate. TC-199—tissue culture medium 199.

**Table 6.** List of other modified media and alternative solution compositions.

| Constituent | Optisol with Supplements [5,23,29,35] | MEM with Supplements [18] | Coconut Water Solution [27] |
|---|---|---|---|
| Base medium | Tissue culture medium 199 Eagle's balance salt solution and MEM | MEM | ACP-412 powdered coconut water |
| Buffer | HEPES | HEPES | Yes |
| Antibiotics | Gentamicin, streptomicin, linezolid and daptomycin | No | Gentamicin 200 µg/mL |
| Chondroitin sulfate | 2.5% | 2.5% | 2.5% |
| Dextran | T-40 | No | 1% |
| ATP precursors | Yes | No | Not reported |
| Sodium bicarbonate | Yes | Yes | Not reported |
| Ascorbic acid | Yes | No | Not reported |
| Sodium pyruvate | Yes | No | Not reported |
| Antifungal additives | Voriconazole or amphotericin B or vancomycin | No | Not reported |
| Additional supplements | α-tocopherol | α-tocopherol | Not reported |

### 3.4. Results of Synthesis

A narrative synthesis was performed, with results grouped according to the type of parameter evaluated by the studies, as shown below.

3.4.1. Cornea Preservation Quantitative Parameters

The most assessed quantitative parameter was the viable endothelial cell density (ECD) (Table 7) employed in nine of the studies [13,22,24,26,27,29–32]. Corneas (769) were evaluated by specular microscopy [13,22,26,32,34]. By assessing this parameter, Basak reported a significant difference for Cornisol at ten days ($p = 0.049$) [22]. Cornea Cold showed statistically significant results from 1 up to 4 weeks, with lower mortality and better preservation of endothelial cells in Cornea Cold when compared with Optisol-GS ($p < 0.05$) [13]. No significant difference was observed for Sinasol versus Optisol-GS for 14 days [28]. Regarding the relationship between time, temperature, and medium composition in Kerasave containing 2.5 mg/mL amphotericin B no statistically significant difference was observed from Optisol-GS up to 14 days at 2–8 °C at all tested time points [30,32]. The ECD in Chen medium was not different between paired corneas at baseline [31].

Despite the Eusol-C medium having a significant difference in composition from Optisol-GS, no statistical difference was found between these solutions up to 7 days [34]. When amphotericin B was tested in Optisol-GS as an antifungal additive in three different concentrations, the decrease in ECD was more significant in the groups supplemented with 0.12-µg/mL amphotericin B for corneas stored between 0–14 days [24]. On the other hand, the decrease in the donor corneal endothelial cell counts after a 7-day storage period in Optisol-GS with the antibiotic daptomycin did not differ significantly from that observed in paired controls stored in unsupplemented Optisol-GS [29].

A total of 266 corneas were evaluated in six studies employing central corneal thickness (CCT) as a measured parameter [13,29–32] (Table 8). There was no statistically significant difference in the CCT values between the Kerasave and Optisol-GS storage groups [30,32]. The use of Cornea Cold resulted in an increase in corneal thickness compared with Optisol-GS at all exposure times (7, 14, 21, and 28 days) [13]. CTT increased in Chen medium (CM) stored corneas after three weeks of storage [31]. As observed with ECD, the inclusion of daptomycin as an additive of the Optisol-GS storage medium resulted in no statistical difference from unsupplemented Optisol-GS (control) during ten days [29].

**Table 7.** Extracted results for endothelial cell density—ECD.

| Author Year | Viable Endothelial Cell Density (ECD) (cells/mm$^2$) * Optisol-GS | Viable Endothelial Cell Density (ECD) (cells/mm$^2$) * Distinct Media |
|---|---|---|
| Basak and Prajna, 2016 [22] | Optisol-GS<br>Day 1: 2868 ± 318<br>Day 3: 2664 ± 352<br>Day 7: 2599 ± 334<br>Day 10: 2482 ± 296<br>Day 14: 2417 ± 384 | Cornisol<br>Day 1: 2731 ± 394; $p = 0.065$<br>Day 3: 2623 ± 356; $p = 0.655$<br>Day 7: 2475 ± 377; $p = 0.177$<br>Day 10: 2349 ± 398; $p = 0.049$<br>Day 14: 2256 ± 336; $p = 0.110$ |
| Duncan et al., 2016 [24] | Optisol-GS mean change in ECD<br>0–14 days, control cornea: 0.06 µg/mL: −191.5; $p = 0.73$<br>0.12 µg/mL: −206.84; $p = 0.01$<br>0.255: −46.75; $p = 0.45$ | Optisol-GS with amphotericin B<br>0.06 µg/mL: 29.92; $p = 0.73$<br>0.12 µg/mL: 114.25; $p = 0.01$<br>0.255: −288.42; $p = 0.45$ |
| Javadi et al., 2021 [26] | Optisol-GS<br>comparison of 7 days:<br>day 1: 2946 ± 457<br>day 7: 2835 ± 493;<br>comparison of 14 days:<br>day 1: 2282 ± 699<br>day 14: 2880 ± 383 | Sinasol<br>comparison of 7 days:<br>day 1: 2829 ± 423; $p = 0.135$<br>day 7: 2723 ± 419; $p = 0.18$;<br>comparison of 14 days:<br>day 1: 2804 ± 423; $p = 0.782$<br>day 14: 2245 ± 589; $p = 0.851$ |
| Kanavi et al., 2015 [27] | Optisol-GS<br>baseline (3151 ± 612); $p = 0.319$<br>1 w (3058 ± 481); $p = 0.319$ | Eusol-C<br>baseline (2925 ± 431); $p = 0.319$<br>1 w (2909 ± 474); $p = 0.319$ |
| Layer et al., 2014 * [28] | Optisol-GS<br>Day 0 to Day 7<br>Voriconazole 50 × MIC: −72.8 (233.7); $p = 0.6$<br>Amphotericin B 0.25 × MIC: −509.9 (497.5); $p = 0.74$<br>Amphotericin B 0.5 × MIC: −107.6 (94.4); $p = 0.27$<br>Amphotericin B 1 × MIC: 84.3 (59.8); $p = 0.07$<br>Amphotericin B 10 × MIC: −99.9 (141.8); $p = 0.04$<br>Day 0 to Day 14<br>Voriconazole 50 × MIC: −182.3 (115.0); $p = 0.41$<br>Amphotericin B 0.25 × MIC: −1026.3 (670.3); $p = 0.38$<br>Amphotericin B 0.5 × MIC: −567.9 (296.3); $p = 0.50$<br>Amphotericin B 1 × MIC: −363.6 (770.8); $p = 0.45$<br>Amphotericin B 10 × MIC: −206.3 (233.9); $p = 0.16$ | Optisol-GS with amphotericin B or voriconazole<br>Day 0 to Day 7<br>Voriconazole 50 × MIC: −4.3 (60.3); $p = 0.6$<br>Amphotericin B 0.25 × MIC: −567.3 (705.4), $p = 0.74$<br>Amphotericin B 0.5 × MIC: 180.9 (243.6); $p = 0.27$<br>Amphotericin B 1 × MIC: −138.4 (140.2); $p = 0.07$<br>Amphotericin B 10 × MIC: indeterminate<br>Day 0 to Day 14<br>Voriconazole 50 × MIC: −259.2 (10.1); $p = 0.41$<br>Amphotericin B 0.25 × MIC: −291.8 (204.4); $p = 0.38$<br>Amphotericin B 0.5 × MIC: −362.7 (102.8); $p = 0.50$<br>Amphotericin B 1 × MIC: −473.1 (288.4); $p = 0.45$<br>Amphotericin B 10 × MIC: indeterminate |

**Table 7.** *Cont.*

| Author Year | Viable Endothelial Cell Density (ECD) (cells/mm$^2$) * Optisol-GS | Viable Endothelial Cell Density (ECD) (cells/mm$^2$) * Distinct Media |
|---|---|---|
| Li et al., 2012 [29] | Optisol-GS<br>Pair 1:<br>without daptomycin: 580 (day 0)<br>without daptomycin: 783 (day 10)<br>Pair 2:<br>without daptomycin: 573 (day 0)<br>without daptomycin: 635 (day 10)<br>Pair 3:<br>without daptomycin: 668 (day 0)<br>without daptomycin: 682 (day 10) | Optisol-GS with linezolid, daptomycin, calcium<br>Pair 1:<br>with daptomycin: 592 (day 0); $p > 0.05$<br>with daptomycin: 715 (day 10); $p > 0.05$<br>Pair 2:<br>with daptomycin: 552 (day 0); $p > 0.05$<br>with daptomycin: 610 (day 10); $p > 0.05$<br>Pair 3:<br>with daptomycin: 649 (day 0); $p > 0.05$<br>with daptomycin: 642 (day 10); $p > 0.05$ |
| Mistò et al., 2020 [30] | Optisol-GS<br>Keratoanalyzer<br>baseline: ECD ranged from 2000 cells/mm$^2$ to 3448 cells/mm$^2$<br>Day 1: (2578 ± 96)<br>Day 7: 2321 ± 145)<br>Day 14: (2335 ± 128)<br>Stocker method<br>Day 1: (2481 ± 71); $p = 0.8974$<br>Day 14: (2050 ± 122); $p = 0.5096$ | Kerasave with amphotericin B and Optisol<br>Keratoanalyzer<br>baseline: ECD ranged from 2000 cells/mm$^2$ to 3448 cells/mm$^2$<br>Day 1: (2521 ± 82); $p = 0.6567$<br>Day 7: (2437 ± 58); $p = 0.4767$<br>Day 14: (2312 ± 98); $p = 0.8863$<br>Stocker method<br>Day 1: (2469 ± 64); $p = 0.8974$<br>Day 14: (2150 ± 87); $p = 0.5096$ |
| Nelson et al., 2000 [31] | Optisol-GS<br>baseline: 2603 ± 356 ($n = 9$); $p = 0.44$<br>Day 21: 2464 ± 243; $p = 0.02$ * ($n = 9$) | Chen medium<br>baseline: 2544 ± 312 ($n = 9$); $p = 0.44$<br>Day 21: 2264 ± 217 ($n = 7$); $p = 0.02$ |
| Parekh et al., 2014 [13] | Optisol-GS<br>baseline av 1900 (1500—2150)<br>1 w loss of 8.03% (± 6.6);<br>2 w loss increased 8.01% (± 6.5);<br>3 w loss 10.99 (± 10.03).<br>4 w loss 16.48 (± 13.5) | Cornea Cold<br>baseline average of<br>1900 (1500–2150) $p = 0.60$<br>1 w loss of 2.94% (± 3.72) in Cornea Cold; $p < 0.05$<br>2 w loss increased<br>4.83% (±5.03); $p < 0.05$<br>3 w loss of 2.66% (±4.44) $p < 0.05$<br>4 w loss of 5.75% (±6.51) $p < 0.05$ |

**Table 7.** *Cont.*

| Author Year | Viable Endothelial Cell Density (ECD) (cells/mm$^2$) * Optisol-GS | Viable Endothelial Cell Density (ECD) (cells/mm$^2$) * Distinct Media |
|---|---|---|
| Perry et al., 2020 [32] | Optisol-GS<br>baseline: 2918 ± 55<br>PK group:<br>Day 1: 3052 ± 66<br>Day 6: 2976 ± 94<br>Day 12: 3060 ± 116<br>DSAEK group:<br>Day 1: 2846 ± 91<br>Day 4: 2847 ± 115<br>Day 6: 2847 ± 71<br>DMEK group:<br>Day 1: 2750 ± 121<br>Day 4: 2779 ± 145<br>Day 6: 2777 ± 140 | * Kerasave<br>baseline: 2887 ± 69; $p$ = 0.734 ($n$ = 22)<br>PK group ($n$ = 10):<br>Day 1: 3096 ± 77; $p$ = 0.669<br>Day 6: 3025 ± 118 ($n$ = 5); $p$ = 0.756<br>Day 12: 2999 ± 132 ($n$ = 5); $p$ = 0.735<br>DSAEK group ($n$ = 7):<br>Day 1: 2740 ± 134; $p$ = 0.527<br>Day 4: 2824 ± 174; ($n$ = 6) $p$ = 0.916<br>Day 6: 2810 ± 173; ($n$ = 6) $p$ = 0.844<br>DMEK group: ($n$ = 5)<br>Day 1: 2677 ± 82; $p$ = 0.632<br>Day 4: 2814 ± 94 ($n$ = 5); $p$ = 0.843<br>Day 6: 2682 ± 74 ($n$ = 5); $p$ = 0.563 |

* Mean (SD) change in endothelial cell count.

**Table 8.** Extracted results for central corneal thickness CCT.

| Author Year | Central Corneal Thickness (CCT) (µm) (Mean ± sd) Optisol-GS | | | | Central Corneal Thickness (CCT) (µm) (Mean ± sd) Distinct Media | | | |
|---|---|---|---|---|---|---|---|---|
| | Days | (n) | CCT | $p$ value | Days | (n) | CCT | $p$ value |
| | Optisol-GS<br>Day 0: Optisol-GS without daptomycin<br>Pair 1 | 2 | 580 (media of pairs) | | Optisol-GS with linezolid, daptomycin, calcium<br>Day 0: Optisol-GS with daptomycin Pair 1 | 2 | 592 | $p$ > 0.05 |
| Li et al., 2012 [29] | Day 10: Pair 1 | 2 | 783 (media of pairs | | Day 10: Pair 1 | 2 | 715 | $p$ > 0.05 |
| | Day 0: Optisol without daptomycin Pair 2 | 2 | 573 | | Day 0: Optisol with daptomycin Pair 2 | 2 | 552 | $p$ > 0.05 |
| | Day 10: Optisol-GS without daptomycin<br>Pair 2 | 2 | 635 | | Day 10: Optisol-GS with daptomycin<br>Pair 2 | 2 | 610 | $p$ > 0.05 |
| | Day 0: Optisol without daptomycin Pair 3 | 2 | 668 | | Day 0: Optisol with daptomycin Pair 3 | 2 | 649 | $p$ > 0.05 |
| | Day 10: Pair 3 | 2 | 682 | | Day 10: Pair 3 | 2 | 642 | $p$ > 0.05 |
| | Day 0: Optisol-GS without daptomycin<br>Pair 1 | 2 | 580 (media of pairs) | | Day 0: Optisol-GS with daptomycin Pair 1 | 2 | 592 | $p$ > 0.05 |

**Table 8.** *Cont.*

| Author Year | Central Corneal Thickness (CCT) (µm) (Mean ± sd) Optisol-GS | | | | Central Corneal Thickness (CCT) (µm) (Mean ± sd) Distinct Media | | | |
|---|---|---|---|---|---|---|---|---|
| Mistò et al., 2020 [30] | Optisol-GS baseline: 551 µm to 740 µm | | | | Kerasave with amphotericin B and Optisol. baseline: 551 µm to 740 µm | | | (*p* = 0.026) |
| | 1 | 16 | (637 ± 10) | | 1 | 16 | (629 ± 13) | 0.6281 |
| | 7 | 16 | (640 ± 19) | | 7 | 16 | (672 ± 15) | 0.1939 |
| | 14 | 16 | (697 ± 19) | | 14 | 16 | (717 ± 17) | 0.4543 |
| Nelson et al., 2000 [31] | Optisol-GS Day 0: | 9 | 0.65 ± 0.06 | | Chen medium Day 0: | 9 | 0.69 ± 0.05 | 0.001 |
| | 7 | 9 | 0.59 ± 0.07 | | 7 | 9 | 0.69 ± 0.06 | 0.0001 |
| | 10 | 6 | 0.63 ± 0.03 | | 10 | 6 | 0.73 ± 0.08 | 0.01 |
| | 14 | 4 | 0.60 ± 0.02 | | 14 | 4 | 0.87 ± 0.04 | 0.0001 |
| | 21 | 2 | 0.69 ± 0.02 | | 21 | 2 | 0.87 ± 0.03 | 0.02 |
| | Day 0: | 9 | 0.65 ± 0.06 | | Day 0: | 9 | 0.69 ± 0.05 | 0.001 |
| Parekh et al., 2014 [13] | Optisol-GS baseline: | | 550 ± 50 | | Cornea Cold baseline: | | 550 ± 50 | |
| | 7 | 12 | 8.5% (±8.0) increase | | 7 | 12 | 6.3% (±8.2) | (*p* < 0.05) |
| | 14 | 12 | 7.8% (±7.5 | | 14 | 12 | 5.3% ± 7.33 | (*p* < 0.05) |
| | 21 | 12 | 6.2% (±7.17); | | 21 | 12 | 3.9% (±5.5) | (*p* < 0.05) |
| | 28 | 12 | 3.84% (±6.15). | | 28 | 12 | 2.86% (±6.28) | (*p* < 0.05) |
| Perry et al., 2020 [32] | Optisol-GS baseline: | 22 | 526 ± 10 | 0.006 | Kerasave baseline: | 22 | 571 ± 12 | 0.006 |
| | Day 1:PK group | | 522 ± 17 | 0.132 | Day 1:PK group | | 563 ± 19 | 0.132 |
| | Day 6: | 5 | 556 ± 30 | 0.311 | Day 6: | 5 | 596 ± 23 | 0.311 |
| | Day 12 | 5 | 594 ± 30 | 0.756 | Day 12 | 5 | 608 ± 32 | 0.756 |
| | DSAEK group: | | 521 ± 18 | 0.161 | DSAEK group: | | 561 ± 20 | 0.161 |
| | Day 4: | 6 | 547 ± 19 | 0.078 | Day 4: | 6 | 600 ± 20 | 0.078 |
| | Day 6: | 6 | 115 ± 9 | 0.784 | Day 6: | 6 | 112 ± 10 | 0.784 |
| | DMEK group: | | | | DMEK group: | | | |
| | Day 4: | 5 | 540 ± 18 | 0.072 | Day 4: | 5 | 602 ± 24 | 0.072 |
| | Day 6: | 5 | not reported | | Day 6: | 5 | not reported | |

The endothelial cell (EC) mortality was assessed in 114 corneas in three studies [22,30,31] (Table 9). A higher EC was observed for Optisol-GS ($p$ = 0.002) when compared with the initial values for corneas treated with Kerasave ($p$ = 0.033) [30]. On Day 14, the extent of mortality was greater in the peripheral area of corneas than in the central area for both storage groups ($p$ = 0.0029 for OGS and $p$ = 0.021 for Kerasave) [30]. Cornisol was comparable with Optisol-GS (17.4% vs. 15.7%; $p$ = 0.83) regarding EC mortality at 14 days [22].

3.4.2. Cornea Preservation Qualitative Parameters

Endothelial cell morphology was assessed in eight studies [13,23,25,26,29–31] by the methods of slit-lamp biomicroscopy [23,26,31], specular microscopy [26], and electron microscopy [20,25,26,31] (Table 10). Three studies assessed the corneal transparency [13,23,30] (Table 10). The complete data characterizing the studies are provided in Tables 4, 9 and 10. Most studies reported no statistical difference in qualitative results (corneal transparency and EC morphology) [29,30]. The comparative study conducted by Parekh et al., 2014 reported that Cornea Cold performed better than Optisol-GS, subjectively, for all parameters ($p$ < 0.05) [13].

**Table 9.** Endothelial cell (EC) mortality or loss.

| Author Year | Endothelial Cell (EC) Mortality (%)<2% Optisol-GS | Endothelial Cell (EC) Mortality (%) <2% Distinct Media | Study Conclusion |
|---|---|---|---|
| Basak and Prajna, 2016 [22] | Optisol-GS<br>Endothelial cell loss (%)<br>Day 3: 7.1 ($n = 31$)<br>Day 7: 9.4 ($n = 30$)<br>Day 10: 13.5 ($n = 27$)<br>Day 14: 15.7 ($n = 19$) | Cornisol<br>Endothelial cell loss (%) $p = 0.1563$<br>Day 3: 3.9; $p = 0.759$<br>Day 7: 9.4; $p = 0.392$<br>Day 10: 14.0; $p = 0.637$<br>Day 14: 17.4; $p = 0.824$ | This study concludes that CS is as effective as OS for storing corneal tissues at 2–8 °C. |
| Mistò et al., 2020 [30] | Optisol-GS<br>%EC mortality ($n = 16$)<br>Central cornea:<br>Day 14: 0.14; $p = 0.0029$<br>Peripheral cornea:<br>Day 1: 0.38 ± 0.20<br>Day 14: 3.38% ± 0.78%; $p = 0.002$ | Kerasave with amphotericin B and Optisol.<br>%EC mortality ($n = 16$)<br>Central cornea: Day 14: 0.54; $p = 0.021$<br>Peripheral cornea:<br>Day 1: 0.56 ± 0.34; $p = 0,97$<br>Day 14: 3.07% ± 0.93%; $p = 0.033$ | Day 14, EC mortality (3.07% ± 0.93% in Kerasave and 3.38% ± 0.78% in Optisol-GS) was higher in both groups as compared to the initial values ($p = 0.033$ for corneas in Kerasave and $p = 0.002$ in Optisol-GS); it was comparable between groups ($p = 0.62$). Day 14, the extent of mortality was higher in the peripheral area of corneas than in the central area for both storage groups ($p = 0.021$ for Kerasave and $p = 0.0029$ for Optisol-GS), whereas it was comparable at the initial evaluation ($p = 0.17$ for Kerasave and $p = 0.15$ for Optisol-GS). |
| Nelson et al., 2000 [31] | Optisol-GS<br>Cell loss (%) ($n = 9$)<br>After 21 days: 5 ± 5 | Chen medium<br>Cell loss (%) ($n = 9$)<br>After 21 days: 11 ± 10; $p = 0.18$ | The two storage media did not differ with respect to endothelial cell loss during storage or to the percentage of TUNEL-positive cells or keratocyte density at the end of the storage period. |

**Table 10.** Endothelial Cell (EC) morphology parameters.

| Author Year | Endothelial Cell (EC) morphology Optisol-GS | Endothelial Cell (EC) Morphology Distinct Media | Study Conclusion |
|---|---|---|---|
| Cid et al., 2021 [23] | Optisol-GS accentuated corneal edema in all groups after 7 days. | Coconut water-based solution accentuated corneal edema in all groups after 7 days. | The preservative solution with coconut water was not effective for human eye bank use. |
| Greenbaum et al., 2004 [25] | Optisol-GS Day 1: All corneas showed progressive epithelial damage. Cells exhibited mild separation and some began to fall off. Some epithelial cells were flattened and all were tightly adherent with a normal appearance of the cytoplasm and nucleus. Day 2: Cells exhibited less defined cell shapes and borders. Sloughing of the external medium and epithelial cell layers. The epithelium was reduced in thickness and was composed generally of two to three cell layers. The basal cell layer appeared normal with intact cell junctions and preserved intracelular organelles; this layer remained attached to the basement membrane. Day 4: All superficial epithelial cell layers were lost and were left with only a basal cell layer, which in some sections showed a mild separation from the basement membrane. Some deep squamous cells sent projections between two adjacent basal cells, reaching the basal membrane, and even separating the basal surface of the cell from the basement membrane. The basal cells did, however, contain normal cellular organelles. | Dexsol Medium Day 1: All corneas showed progressive epithelial damage. Cells exhibited mild separation and some began to fall off. Some epithelial cells were flattened and all were tightly adherent with a normal appearance of the cytoplasm and nucleus. Day 2: Cells exhibited less defined cell shapes and borders. Sloughing of the external medium and epithelial cell layers. The epithelium was reduced in thickness and was composed generally of two to three cell layers. The basal cell layer appeared to be normal with intact cell junctions and preserved intracelular organelles; this layer remained attached to the basement membrane. Day 4: All superficial epithelial cell layers were lost and were left with only a basal cell layer, which in some sections showed a mild separation from the basement membrane. Some deep squamous cells sent projections between two adjacent basal cells, reaching the basal membrane and even separating the basal surface of the cell from the basement membrane. The basal cells did, however, contain normal cellular organelles. | According to this study, Optisol appears to be no more effective than Dexsol in preserving the integrity of human corneal epithelium. |

| Author Year | Endothelial Cell (EC) morphology Optisol-GS | Endothelial Cell (EC) Morphology Distinct Media | Study Conclusion |
|---|---|---|---|
| Javadi et al., 2021 [26] | Optisol-GS:<br>Day 1:<br>stromal edema<br>not mild: 28 (100%); $p > 0.999$<br>moderately severe: 0 (0.0%); $p > 0.999$<br>Descemet's folding:<br>not mild: 27 (96.4%); $p = 0.038$<br>moderately severe: 1 (3.6%); $p = 0.038$<br>Day 14:<br>stromal edemanot mild: 12 (42.9%); $p > 0.999$<br>moderately severe: 16 (57.1%); $p > 0.999$<br>Descemet's folding:not mild: 0 (0.0%); $p > 0.999$<br>moderately severe: 28 (100.0%); $p > 0.999$ | Sinasol:<br>Day 1:<br>stromal edema<br>not mild: 30 (96.8%); $p > 0.999$<br>moderately severe: 1 (3.2%); $p > 0.999$<br>Descemet's folding:<br>not mild: 23 (74.2%); $p = 0.038$<br>moderately severe: 8 (25.8%); $p = 0.038$<br>Day 14:<br>stromal edema<br>not mild: 10 (32.3%); $p = 0.488$<br>moderately severe: 21 (67.7%); $p = 0.488$<br>Descemet's folding:<br>not mild: 0 (0.0%); $p > 0.999$<br>moderately severe: 31 (100.0%); $p > 0.999$ | Our study demonstrated no significant difference in the mean area of dead ECs and denuded Descemet's membrane between the two storage media after 7- and 14-day periods. |
| Kanavi et al., 2015 [27] | Optisol-GS<br>stromal edema:<br>Day 1:<br>no edema: 85 (100.0%); $p = 0.554$<br>mild edema: 0 (0.0%); $p = 0.554$<br>mod. edema: 0 (0.0%); $p = 0.554$<br>severe edema: 0 (0.0%); $p = 0.554$<br>Day 7: no edema: 83 (97.6%); $p = 0.554$<br>mild edema: 2 (2.4%); $p = 0.554$<br>mod. edema: 0 (0.0%); $p = 0.554$<br>severe edema: 0 (0.0%); $p = 0.554$<br>Descemet's folding:<br>Day 1:<br>significant DF: 0 (0.0%); $p = 0.325$<br>non-significant DF: 85 (100.0%); $p = 0.325$<br>significant vac: 10 (11.8%); $p = 0.687$<br>non-significant vac: 75 (88.2%); $p = 0.687$<br>Day 7: significant DF: 4 (4.7%); $p = 0.325$<br>non-significant DF: 81 (95.3%); $p = 0.325$<br>significant vac: 30 (35.3%); $p = 0.687$<br>non-significant vac: 55 (64.7%); $p = 0.687$ | Eusol-C<br>stromal edema:<br>Day 1:<br>no edema: 86 (100.0%); $p = 0.554$<br>mild edema: 0 (0.0%); $p = 0.554$<br>mod. edema: 0 (0.0%); $p = 0.554$<br>severe edema: 0 (0.0%); $p = 0.554$<br>Day 7: no edema: 85 (98.8%); $p = 0.554$<br>mild edema: 1 (1.2%); $p = 0.554$<br>mod. edema: 0 (0.0%); $p = 0.554$<br>severe edema: 0 (0.0%); $p = 0.554$<br>Descemet's folding:<br>Day 1:<br>significant DF: 0 (0.0%); $p = 0.325$<br>non-significant DF: 86 (100.0%); $p = 0.325$<br>significant vac: 16 (18.6%); $p = 0.687$<br>non-significant vac: 70 (81.4%); $p = 0.687$<br>Day 7: significant DF: 19 (22.1%); $p = 0.325$<br>non-significant DF: 67 (77.9%); $p = 0.325$<br>significant vac: 31 (36.9%); $p = 0.687$<br>non-significant vac: 53 (63.1%); $p = 0.687$ | In conclusion, the changes of overall cornea rating, endothelial cell indices, stromal edema, and Descemet's folding from Day 1 to Day 7 were not significantly different between Optisol-GS and Eusol-C and none of them seem to be superior to another. |

**Table 10.** *Cont.*

| Author Year | Endothelial Cell (EC) morphology Optisol-GS | Endothelial Cell (EC) Morphology Distinct Media | Study Conclusion |
|---|---|---|---|
| Mistò et al., 2020 [30] | Optisol-GS: polymorphism: mild/medium and altered, *n* Day 1: 10/6 (16)); *p* < 0.05 Day 14: 5/10 (15); *p* < 0.05 endothelial cell borders homogeneous and partly homogeneous/irregular and altered, *n* Day 1: 5/11 (16); *p* < 0.05 Day 14: 1/12 (13); *p* < 0.05 | Kerasave: polymorphism: mild/medium and altered, *n* Day 1: 11/5 (16); *p* < 0.05 Day 14: 5/10 (15); *p* < 0.05 endothelial cell borders homogeneous and partly homogeneous/irregular and altered, *n* Day 1: 5/11 (16); *p* < 0.05 Day 14: 0/14 (14); *p* < 0.05 | No significant differences were found in all corneal gradings between Day 1 and Day 14 (all *p* values > 0.05). Ten corneas stored in Optisol-GS (out of 16 = 62.5%) and five in Kerasave (out of 16 = 33.3%) changed transparency grade over time without significant difference. Both groups showed increased polymorphism on Day 14, yet without statistically significant difference between groups. In both groups, EC borders showed an increase in irregularities after 14-day storage, without significant difference between groups. |
| Nelson et al., 2000 [31] | Optisol-GS baseline: coefficient of variation of cell size: 0,27 ± 0.04; *p* = 0.02 hexagonal cell (%): 64 ± 9; *p* = 0.02 Day 14: coefficient of variation of cell size: 0.31 ± 0.05; *p* = 0.65 hexagonal cell (%): 63 ± 9; *p* = 0.48 | Chen medium baseline: coefficient of variation of cell size: 0.28 ± 0.05; *p* = 0.65 hexagonal cell (%): 63 ± 8; *p* = 0.48 Day 14: coefficient of variation of cell size: 0.32 ± 0.05; *p* = 0.65 hexagonal cell (%): 62 ± 4; *p* = 0.48 | Corneas stored in CM were thicker during storage than those stored in OGS. SEM of the endothelium of four paired corneas after storage in CM and OGS. The endothelium appears intact in all corneas. The abnormal appearance of the endothelium after storage in CM for 14 and 21 days may be related to the dramatic increase in stromal swelling. |
| Parekh et al., 2014 [13] | Optisol-GS low level of cells: 1 w: 1.6% (±10.4) with statistical significance (*p* < 0.05) 2 w: 4.5% (±10.7) (*p* < 0.05) 3 w: 5.4% (±8.3) (*p* < 0.05) 4 w: 4.38% (±9.8) (*p* < 0.05) | Cornea Cold medium low level of cells: 1 w: 14.5% (±14.86) with statistical significance (*p* < 0.05) 2 w: 8.9% (±11.2) (*p* < 0.05) 3 w: 12.9% (±15.6) (*p* < 0.05) 4 w: 14.75 (±15.2) (*p* < 0.05) | Cornea Cold is a promising hypothermic corneal storage medium with preservation time until 21 days. |

**Table 10.** *Cont.*

| Author Year | Endothelial Cell (EC) morphology Optisol-GS | Endothelial Cell (EC) Morphology Distinct Media | Study Conclusion |
|---|---|---|---|
| Tachibana and Sawa, 2002 [20] | Optisol-GS Day 7: in cornea stored in Optisol-GS, SEM revealed distinctive corneal endothelial cell boundaries and TEM showed almost normal findings in intracellular organelles. Day 14: In cornea stored in Optisol-GS or Medium l, cell boundaries became blurred in several places and vacuoles in the cytoplasm were observed. These findings indicated that corneal endothelial cells were well preserved in their morphological aspects. | MEM with 2.5% chondroitin sulfate with different molecular weights in Medium I and Medium II Day 7: in cornea stored in Medium I, SEM revealed distinctive corneal endothelial cell boundaries and TEM showed almost normal findings in intracellular organelles. Day 14: Medium I could be similarly potent and maintain morphological characteristics as well as Optisol-GS in a 14-day preservation period. Medium II Day 7: cornea in Medium II showed a paving-stone-like bulging of endothelial cell surface but had almost normal intracellular organelles.Day 14: Cornea in Medium II demonstrated irregularly shrunken cell surfaces by SEM examination, and the degenerated cells showed less staining pattern, destroyed cristae in mitochondria, and aggregation of intranuclear chromatin by TEM examination. Cornea stored in Medium II showed greater deterioration than the corneas stored in the other two media. | In the present morphological study, a novel formulated solution with simple ingredients showed a capability for corneal preservation similar to Optisol-GS. |

## 4. Discussion

Nowadays, Optisol-GS remains the most widely employed commercial solution for tissue preservation aimed at cornea transplantation worldwide. Eye banking documents recommend using an appropriate corneal storage solution following good manufacturing practices, which shall be used and stored according to the manufacturer's recommendations for temperature and time limit [35,37]. The differences between this gold standard product and other available solutions have been the object of research in several studies. These studies show methodological differences in their evaluation protocols, ranging from very diverse numbers of corneas used to different preservation times, temperature range, and composition of preservative solutions. Nevertheless, most studies do not identify relevant differences between Optisol-GS and other products, as assessed by both qualitative and quantitative parameters, in the usual 7-day period of preservation common to cornea banking. However, the literature presents some evidence of differences, especially in longer times, that may be related to the chemical composition of the preservation media, as will be discussed below.

One of the factors that may contribute to the preservation and reduction of the rate of vacuolization of endothelial cells in the Optisol-GS medium is the presence of chondroitin sulfate, non-essential amino acids, ascorbic acid, vitamins, purines, and lipid components with antioxidant action. These components were initially advertised, along with an increased concentration of chondroitin sulfate, as improvements that should provide superior preservative properties when compared with Dexsol, an older preservative medium available in the USA [25]. However, the study of Greenbaum et al. [25] reported very similar performances between Dexsol and Optisol-GS. However, the authors observed a loss of epithelium that was related to the conservation time and suggested that the preservation time of corneas should be lower than four days, especially when performing penetrating keratoplasty in patients with alterations of the ocular surface. The authors stated that efforts must continue to find an optimal corneal storage medium that is cost effective, safe, and ensures minimal loss of epithelial cells [25].

Cornisol is an intermediate-term corneal storage medium developed in India, aimed at improving the affordability and availability of preservation media for keratoplasty in developing countries. It differs from Optisol GS by adding insulin, vitamins, coenzymes, and trace elements. Cornisol is available only outside the US and approved in India for cold storage up to 14 days. A single study conducted by Basak and Prajna [22] compared Cornisol and Optisol GS, reporting the absence of differences in 14 days regarding the endothelial cell loss, endothelial cell density (ECD), and percentage of hexagonality evaluation. The fact that that Cornisol costs usually half the price of Optisol-GS [22] is one important feature of this formulation if further studies confirm similar clinical outcomes after transplantations performed with corneas preserved with both media.

The studies by Tachibana et al. [20,33] discuss the requirement of intermediate corneal storage in Japan since the government has regulated the quarantine time of donors to avoid infectious diseases. This requirement raised the need to develop a domestic composition with a simple formulation for longer preservation, which was investigated for safety reasons in rabbit corneas. The authors proposed in 2002 a new solution, whose main differences from Optisol GS are the absence of dextran, streptomycin, and ATP sources [33]. This new solution was formulated with a simple combination of MEM with 2.5% chondroitin sulfate. This formulation showed a similar corneal preservation ability as that presented by Optisol-GS in both rabbit [20] and human corneas [33]. The authors proposed that the maintenance of cornea transparency and the prevention of stromal edema are related to the high osmolarity of the solutions, which was around 365 mOsm/kg in Optisol-GS.

The Chen medium (CM) differs from Optisol-GS mainly by the presence of β-hydroxybutyrate in its composition, a ketoacid that has been found to stimulate corneal endothelial cell proliferation. Yap et al. [38] compared CM and Optisol-GS stored corneas for 48 h, reporting no significant difference in the corneal thickness. However, this study investigated a relatively short time of preservation, limiting the generalization to the needs

of most eye banks. Later, Nelson et al. [31] observed no difference in corneal thickness from baseline in CM stored corneas after one week of storage, but this parameter increased dramatically from 1 to 2 weeks of storage [31,38]. While longer preservation times need to be investigated to appropriately compare the adequacy of CM compared with Optisol-GS, two different clinical trials have identified very similar clinical outcomes after using Chen medium or Optisol-GS in penetrating keratoplasties with preservation under 7 days [39,40].

Presently, Optisol-GS does not include any antifungal additive. In addition, its colorimetric indicator, phenol red, does not reliably detect *C. albicans* contamination. The Eye Bank Association of America (EBAA) medical advisory board did not recommend antifungal supplementation of corneal storage media, partly because of insufficient evidence regarding efficacy and safety. In this context, Layer et al. [28] investigated the inclusion of voriconazole and amphotericin B as additives to Optisol-GS in reducing *C. albicans* and *C. glabrata* contamination under normal storage conditions. Their findings suggest that a low concentration of amphotericin B might be a safe and efficacious addition to storage media, even though more extensive studies are necessary to confirm these data [41]. In this sense, another medium containing an antimycotic tablet is Kerasave, which presented 2.5 µg/mL of amphotericin B and which was recently studied by Mistó et al. [30], in a comparison that evidenced properties comparable with Optisol-GS in terms of corneal preservation at 2–8 °C for 14 days. The final approval by the FDA for Kerasave is still pending, as it has been delayed during the COVID-19 pandemic.

As an important exception, Cornea Cold was the cold storage medium that presented better results in all qualitative parameters and tests compared with Optisol GS, from 1 to 14 days. According to Parekh et al. [13], the usual assessments employed to evaluate preservation media are subjective since there is still no standardized protocol for corneal preservation at eye banks, making it difficult to recommend a specific media for conservation before surgery. Therefore, the authors propose the standardization of the evaluation of cornea quality through a quantitative overall quality (OQ) assessment that integrates different comparative parameters such as thickness, transparency, viable endothelial cell density (VECD), and morphology on a four-point numerical scale, including longer preservation times up to 28 days, widely used in the technique of organ culture at 31 °C [13]. With such methodology, the authors identified similar OQ rates for Cornea Cold with a one-week delay compared with Optisol-GS and, therefore, proposed a maximum storage time of 21 days. Such a longer preservation time proposed for Cornea Cold could theoretically be beneficial for long-route international transportation of surplus corneas to countries with higher demand for human corneal tissues. The authors also propose that the main advantage of Cornea Cold is the maintenance of thinner tissues with high transparency, which could allow for better observation and manipulation for earlier visual rehabilitation. However, such proposals must be considered with caution since this observation is limited to a single study with a relatively low number of samples, requiring further studies and confirmation for greater certainty of the evidence.

Eusol-C, one of the main preservation media used in Europe, presents a similar composition of Cornea Cold, including a high concentration of dextran and the absence of ascorbic acid and chondroitin sulfate. Dextran is an osmotically active agent present in most commercial preservation preparations that penetrates the cornea during preservation [36]. Despite the similar composition between Eusol-C and Cornea Cold, these media performed differently when compared with Optisol-GS, as no statistical difference was found between Eusol-C and Optisol-GS solutions up to 7 days [25,42]. Camposampiero et al. [42] reported that the endothelial cell density of the corneas preserved for one week in Optisol-GS was the same as that of the corneas kept in Eusol-C for 1–2 days. Nevertheless, these results cannot be interpreted directly due to a lack of difference between the preservative preparations since the comparisons were not carried out under similar time conditions. Furthermore, other quantitative and qualitative endpoints related to cornea integrity were not evaluated [42]. The presence of 2 mM glutamine as a basic supplement in the Eusol-C medium, was proposed as compensating for the absence of the additives present in Optisol

GS, causing the same final effect on cornea preservation [43]. The findings reported by Kanavi et al. [27] suggest that, in eye banks that operate with a short interval between preservation and transplantation (7 days), Eusol-C would be a proper substitute for Optisol-GS. A recent issue has brought attention to the relevance of comparing the performance of Optisol-GS and Eusol-C, as a shortage of Optisol in the US reported by the American Academy of Ophthalmology by April 2022 [44], due to supply chain difficulties, led to the Eye Bank Association of America (EBAA) arranging for the importation of Eusol-C, mainly marketed in Europe. In this context, the literature provides evidence of similar performances on donor tissue preservation for keratoplasty [27,42]. Furthermore, Eusol has been reported as presenting lower operational costs outside of the US [27].

One of the main limitations of the literature assessed in the present review was the restricted number of corneas enrolled in the selected studies due to a high demand for donor corneas and short preservation-to-transplantation times [27]. In addition, there are probably other available outcomes of the use of different cold corneal storage media compared with Optisol-GS that may not have been assessed in this review. Preliminarily, the intent of this review included the conduction of meta-analyses comparing such results, depending on the methodological heterogeneity across studies, by calculating the mean difference or relative risk corresponding to the outcomes reported as continuous or categorical data, respectively. However, the assessment of the available data showed the impossibility of this kind of analysis due to: (i) a considerable methodological heterogeneity between the studies, mainly related to differences in the composition of the media compared with Optisol-GS, preservation conditions, and follow-up periods; and (ii) the variation in the type of data reported for the outcomes assessed.

Therefore, this review focused on the mainly employed corneal integrity parameters, for which more data are available to establish comparisons between studies. Regarding the limitations of the review process, it is also important to point out that four papers could not be retrieved as full texts and there is a risk that they could become eligible for this review. Nevertheless, the search strategy employed hereby may have been sufficient to reach a representative sample of the available literature, helping to map and identify the primary evidence for the performance of several different preservation media compared with the gold standard Optisol-GS solution. However, as stated previously, the correlation of data from different studies might have been impaired by the very diverse methodologies employed regarding experimental times, endpoints, and preservation conditions. Consequently, future developments in the field should start with initiatives to standardize the quality of preserved corneas to allow for the homogeneity of findings and comparison of results in the light of similar technical performances. In this sense, the overall quality parameters proposed by Parekh et al. [13] could provide an interesting initial framework for standardization. Further advances in the field may include the use of other contemporary antibiotics and antifungals to ensure longer preservations, along with the use of biological mediators such as fibroblast growth factor 2 (FGF-2) and inhibitors of nitric oxide synthase 2, which are suggested to impact the health of stored cornea [11]. Finally, future cost-benefit studies should be performed regarding the use of the different cold cornea storage media, identifying the advantages of choosing the best medium for a national transplantation program, and considering products with similar technical performances but with lower costs.

## 5. Conclusions

This systematic review identified several different available commercial and non-commercial preservation media compared with Optisol-GS by different endpoints to assess corneal preservation quality using distinct protocols. According to the systematized data of this study, we can conclude that the preservative solution with coconut water was not effective for use in human eye banks. Corneas stored in the Chen medium were thicker during storage than those stored in Optisol-GS. Optisol-GS appears to be no more effective than Kerasave found in all corneal gradings between Day 1 and Day 14, and Dexsol

similarly preserves the integrity of the human corneal epithelium. The changes of overall cornea rating, endothelial cell indices, stromal edema, and Descemet's folding from Day 1 to Day 7 were not significantly different between Optisol-GS and Eusol-C. MEM with 2.5% chondroitin in the present morphological study, a new solution formulated with simple ingredients, showed a corneal preservation capacity similar to Optisol-GS. Cornea Cold is a promising hypothermic corneal storage medium with a preservation time of up to 21 days.

A comprehensive analysis of the composition of the cold storage media revealed important differences. However, most of the selected cold cornea media presented similar effects to Optisol-GS on the preservation of cornea at seven days, while at ten days Cornea Cold[TM] presented better results in one study. While these in vitro results should be carefully correlated to clinical outcomes due to inherent methodological limitations, the similar performance of cornea preservation at seven days indicates that the choice of cold storage media might be motivated by other relevant factors such as availability and operational costs for a transplantation program. Furthermore, the data suggest that the ideal formulation for preservation at longer preservation times remains to be identified, even with some promising results already reported for formulations such as Cornea Cold.

**Supplementary Materials:** The following supporting information can be downloaded at: https://www.mdpi.com/article/10.3390/app12147079/s1. Table S1: PRISMA 2020 Main Checklist.

**Author Contributions:** Conceptualization, I.G. and G.G.A.; methodology, I.G., A.V.B.P., M.d.S.S., G.A.M.-V., L.C.M. and G.G.A.; software, I.G., A.V.B.P. and M.d.S.S.; validation, I.G., A.V.B.P., M.d.S.S. and G.G.A.; formal analysis, I.G., A.V.B.P., M.d.S.S., G.A.M.-V., L.C.M. and G.G.A.; investigation, I.G., A.V.B.P. and M.d.S.S.; resources, I.G.; data curation, I.G., A.V.B.P., M.d.S.S. and G.A.M.-V.; writing—original draft preparation, I.G., A.V.B.P., M.d.S.S. and G.A.M.-V.; writing—review and editing, I.G., A.V.B.P., M.d.S.S., G.A.M.-V., O.A.F.P., M.S.G., L.C.M. and G.G.A.; visualization, I.G., A.V.B.P., M.d.S.S., G.A.M.-V., O.A.F.P., M.S.G., L.C.M. and G.G.A.; supervision, L.C.M. and G.G.A.; project administration, I.G. and G.G.A.; funding acquisition, I.G. All authors have read and agreed to the published version of the manuscript.

**Funding:** This research was funded by INOVA FIOCRUZ, Rio de Janeiro, Brazil, grant number "VPPIS-004-FIO-18-50".

**Institutional Review Board Statement:** Not applicable.

**Informed Consent Statement:** Not applicable.

**Data Availability Statement:** The study protocol was registered in the Open Science Framework Database, available at the link: osf.io/qh69k/.

**Acknowledgments:** The authors gratefully acknowledge the support of INOVA FIOCRUZ, Rio de Janeiro, Brazil, without which the present study could not have been completed. The authors thank the experts Mehta, Reza Dana, and Namrata Sharma for answering the email contacts.

**Conflicts of Interest:** The authors declare no conflict of interest.

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
