# Peer review of "Cold Storage Media versus Optisol-GS in the Preservation of Corneal Quality for Keratoplasty: A Systematic Review"

_applsci, doi:10.3390/app12147079_

Round 1

Reviewer 1 Report

The proponents delved into the use of antibiotics to preserve the quality of cornea. It is recommended that the proponents look into the format of the manuscript as it does not adhere to the Journal of Applied Sciences. The rationale of the paper has to be clearly stated as it does not seem address an unmet clinical need. The proponents are requested to carefully restructure the paper before it can be deemed publishable. 

Author Response

Dear Editors and Reviewers of Applied Sciences

The authors would like to acknowledge the effective and unbiased review of the manuscript, and believe that, by assessing the suggested alterations, produced a manuscript with improved editorial and scientific quality, which we hope is adequate for publishing at Applied Sciences. The authors thank the opportunity of resubmission of a revised manuscript.

All changes made according to the reviewers' and editors' comments were highlighted in red font in the revised manuscript. Please find below the point-by-point answers to the editors’ and reviewers’ comments.

Reviewer 1

  1. Extensive editing of English language and style required

Answer: English language editing was performed with professional help. The certificate was attached to the submission.

  1. The proponents delved into the use of antibiotics to preserve the quality of cornea. It is recommended that the proponents look into the format of the manuscript as it does not adhere to the Journal of Applied Sciences.

Answer: The manuscript was thoroughly revised to adhere to the format of Applied Sciences.

  1. The rationale of the paper has to be clearly stated as it does not seem address an unmet clinical need. The proponents are requested to carefully restructure the paper before it can be deemed publishable. 

Answer: The manuscript was completely revised to (i) better connect the relationship between the choice of an adequate cornea preservation solution and clinical success after the performance of keratinoplasty; (ii) improve the discussion on the relationship between chemical composition and advances on cornea preservation; (iii) provide more assertive conclusions based on the gathered data.

We thank the reviewer for the consideration and contribution for an improved manuscript.

Reviewer 2 Report

The authors present the systematic review paper on “Cold storage media versus Optisol-GS for corneal quality preservation. A systematic review”.

The authors provide a clear methodology in terms of the search strategy for databases, registers, and websites, including any filters and limits used.

This systematic review was reported following the recommendations of the Pre Reporting Items for Systematic Reviews and Meta-Analyses (PRISMA).

The main conclusion is that the evidence indicates that most cold corneal storage media (CCSM) present similar performances on cornea preservation for transplantation at seven days. There is no clear indication to recommend a specific CCSM since all products have similar effects on the cornea.

I have no objections to the methodology of this study, although, in my opinion, the study does not provide any interesting information. All CCSMs are approved for use by the relevant regulatory authorities, so it is obvious that they are safe and adequately effective.

Author Response

Reviewer 2

The authors present the systematic review paper on “Cold storage media versus Optisol-GS for corneal quality preservation. A systematic review”.

The authors provide a clear methodology in terms of the search strategy for databases, registers, and websites, including any filters and limits used.

This systematic review was reported following the recommendations of the Pre Reporting Items for Systematic Reviews and Meta-Analyses (PRISMA).

The main conclusion is that the evidence indicates that most cold corneal storage media (CCSM) present similar performances on cornea preservation for transplantation at seven days. There is no clear indication to recommend a specific CCSM since all products have similar effects on the cornea.

I have no objections to the methodology of this study, although, in my opinion, the study does not provide any interesting information. All CCSMs are approved for use by the relevant regulatory authorities, so it is obvious that they are safe and adequately effective.

Answer: As adequately pointed by the reviewer, most (but not all) identified cold storage media are widely approved by regulatory authorities around the World. Nevertheless, even with minimal insurance of safety and effectivity, these media were not expected, by assumption, to present the same performance, especially in longer preservation times. Since they present very different compositions, costs and, sometimes are advertised as superior to other options in the market, several researchers were motivated to perform the comparisons issued in the manuscript. The main relevance of this review is to compile the results from these comparisons, using the gold-standard medium as a reference, and provide the best evidence for of decision-makers in this important factor for a successful keratoplasty, and cost-effective transplantation programs.

The authors understand, after the comments of the reviewers, that the original manuscript failed to show the relevance of the research question and of the data retrieved from the different studies, including a conclusion that do not completely represents the findings. Therefore, the manuscript was thoroughly revised to provide more relevance to the research question, the relationship between chemical composition and advances on cornea preservation and provide more assertive conclusions based on the gathered data.

Regarding the identification of similar performances for different media, it is important to notice that this result itself may be of great relevance, since some media may present other advantages related to availability and costs, which are now addressed in the revised manuscript. Nevertheless, the revised manuscript now reinforces the differences found between Optisol-GS and other media such as Cornea Cold, which presented a better performance at longer preservation times.

We thank the reviewer for the consideration and contribution for an improved manuscript.

Reviewer 3 Report

·        Author presented a systematic literature review on Cold storage media versus Optisol-GS and investigated the effects of using different cold corneal storage media (CCSM) compared to Optisol-GS on the quality of stored corneas.

·        Summarize your research in one paragraph that shows your paper contribution. Add this summary as a second last paragraph in the introduction section.

·        Way of referencing is not as per journal format. Please correct all your references in one standard format. Reviewer has noticed that some references are in IEEE style and some are in APA (tables). Please make all in one style.

·        Paper organization section is missing in the end of introduction of section. Briefly describe the section and subsection of your whole menu script in one paragraph. Add this paragraph in the end of the introduction section.

·        The defined search string is not working properly as it was applied in your chosen databases. Please modify the search string in order to get relevant results.

·        It has been noticed the sentence length in some section is too long. Please shorter them for a clear understanding of reader.

·        Exclusion and inclusion criteria has not been defined in methodology section. There is no screening process presented. Add screening based on keywording and abstract of fetched articles.

·        Add paragraph between two headings such as in section 3 there should be some text between Results heading and study selection sub heading and same for entire menuscript.

·        Research questions are the main concern any systematic literature review. There is no research questions table presented in this SLR. Make at least 4 research questions with their corresponding motivation. After that address your defined research questions in tabular form. Furthermore, explanation of each question also necessary. Also address the year of selected studies for your research questions.

·        What are challenges and gaps please address

·        Conclusion section is very short brief all your research and make it little more comprehensive.

·        Furthermore, no contribution has been presented in this research. Highlight your contribution or propose any taxonomy, framework, or architecture in order to show the strength as well as contribution of your research.

Author Response

Reviewer 3

Author presented a systematic literature review on Cold storage media versus Optisol-GS and investigated the effects of using different cold corneal storage media (CCSM) compared to Optisol-GS on the quality of stored corneas.

  1. Summarize your research in one paragraph that shows your paper contribution. Add this summary as a second last paragraph in the introduction section.

Answer: a summary of the research and contribution was added near the end of the Introduction, as requested.

  1. Way of referencing is not as per journal format. Please correct all your references in one standard format. Reviewer has noticed that some references are in IEEE style and some are in APA (tables). Please make all in one style.

Answer: the references were reformatted to the journal recommended style (ACS).

  1. Paper organization section is missing in the end of introduction of section. Briefly describe the section and subsection of your whole menu script in one paragraph. Add this paragraph in the end of the introduction section.

Answer: the end of the Introduction section now contains a brief paragraph on the paper organization, as requested.

  1. The defined search string is not working properly as it was applied in your chosen databases. Please modify the search string in order to get relevant results.

Answer: The entire search key has been checked against all databases and formatted in the table 2 of the revised manuscript.

  1. It has been noticed the sentence length in some section is too long. Please shorter them for a clear understanding of reader.

Answer: English language editing was performed with professional help. The certificate was attached to the submission.

  1. Exclusion and inclusion criteria has not been defined in methodology section. There is no screening process presented. Add screening based on keywording and abstract of fetched articles.

Answer: The exclusion and inclusion criteria were specified and add competed information in eligibility criteria (2.2. item of the revised manuscript). The screening processes are presented in the Figure 1, with the PRISMA 2020 Flow diagram of the screening and selection process.

  1. Add paragraph between two headings such as in section 3 there should be some text between Results heading and study selection sub heading and same for entire menuscript.

Answer: The manuscript was corrected accordingly.

  1. Research questions are the main concern any systematic literature review. There is no research questions table presented in this SLR. Make at least 4 research questions with their corresponding motivation. After that address your defined research questions in tabular form. Furthermore, explanation of each question also necessary. Also address the year of selected studies for your research questions.

Answer: The section 2.2 and the Table 1 of the revised manuscript now present the four research questions that composed the main research question, according to the PICO framework, following the recommendations of the PRISMA Statement and the Cochrane Handbook for Systematic Reviews of Interventions.

  1. What are challenges and gaps please address

Answer: The revised discussion was improved to include the main challenges and gaps in its last paragraphs, as well as future perspectives of research, and limitations of the literature (and of the review process).

  1. Furthermore, no contribution has been presented in this research. Highlight your contribution or propose any taxonomy, framework, or architecture in order to show the strength as well as contribution of your research.

Answer: There are several different cold storage media for the preservation of cornea for keratoplasty, and there is evidence that this preservation is a risk factor for the clinical success of cornea transplantation. The media present very different compositions, costs and, sometimes are advertised as superior to other options in the market, motivating several researchers to perform the comparisons issued in the manuscript. The main contribution of the manuscript is to systematically compile and present these findings, using the gold-standard medium as a reference, and provide the best evidence for of decision-makers in this important factor for a successful keratoplasty, and cost-effective transplantation programs.

The authors recognize that the original version of the manuscript failed to highlight this contribution and, therefore, the manuscript was thoroughly revised to (i) better connect the relationship between the choice of an adequate cornea preservation solution and clinical success after the performance of keratinoplasty; (ii) improve the discussion on the relationship between chemical composition and advances on cornea preservation; (iii) provide more assertive conclusions based on the gathered data.

We thank the reviewer for the consideration and contributions for an improved manuscript.

Round 2

Reviewer 1 Report

The proponents delved into a review of the Cold storage media versus Optisol-GS in the preservation of corneal quality for keratoplasty. The paper has substantially improved compared with the initial submission. There are minor formatting issues in the manuscript. 

Author Response

Dear Editors and Reviewers of Applied Sciences

The authors would like to acknowledge the effective and unbiased review of the manuscript, and believe that, by assessing the suggested alterations, produced a manuscript with improved editorial and scientific quality, which we hope is adequate for publishing at Applied Sciences. The authors thank the opportunity of resubmission of a revised manuscript.

All changes made according to the reviewers' and editors' comments were highlighted in red font in the revised manuscript. Please find below the point-by-point answers to the editors’ and reviewers’ comments.

Reviewer 1

  1. Moderate English changes required

Answer: English language editing was performed with professional help. The certificate was attached to the submission.

  1. The proponents delved into a review of the Cold storage media versus Optisol-GS in the preservation of corneal quality for keratoplasty. The paper has substantially improved compared with the initial submission. There are minor formatting issues in the manuscript.

Answer: The manuscript was thoroughly revised to adhere to the format of Applied Sciences. In this sense, Author List and Affiliations were corrected; Abstract word count was revised and adjusted; a paper outline regarding methods, results, discussion and conclusion sections was inserted in the introduction; Table titles were formatted; and Reference numbers were formatted according to the journal's rules.

We thank the reviewer for the consideration and contribution for an improved manuscript.

This manuscript is a resubmission of an earlier submission. The following is a list of the peer review reports and author responses from that submission.